# Argue with Me Tersely: Towards Sentence-Level Counter-Argument Generation

**Jiayu Lin[1], Rong Ye[1,2], Meng Han[3], Qi Zhang[1],**
**Ruofei Lai[3], Xinyu Zhang[3], Zhao Cao[3], Xuanjing Huang[1], Zhongyu Wei[1*]**
[1]Fudan University, [2]ByteDance, [3]Huawei Poisson Lab
jiayulin22@m.fudan.edu.cn,yerong@bytedance.com
{qz,xjhuang,zywei}@fudan.edu.cn
{hanmeng12,lairuofei,zhangxinyu35,caozhao1}@huawei.com

## Abstract

Counter-argument generation—a captivating area in computational linguistics—seeks to craft statements that offer opposing views. While most research has ventured into paragraph-level generation, sentence-level counter-argument generation beckons with its unique constraints and brevity-focused challenges. Furthermore, the diverse nature of counter-arguments poses challenges for evaluating model performance solely based on n-gram-based metrics. In this paper, we present the **ArgTersely** benchmark for sentence-level counter-argument generation, drawing from a manually annotated dataset from the Change-MyView debate forum[1]. We also propose **Arg-LlaMA** for generating high-quality counter-argument. For better evaluation, we trained a BERT-based evaluator **Arg-Judge** with human preference data. We conducted comparative experiments involving various baselines such as LlaMA, Alpaca, GPT-3, and others. The results show the competitiveness of our proposed framework and evaluator in counter-argument generation tasks. Code and data are available at `https://github.com/amazingljy1206/ArgTersely`.

## 1 Introduction

Counter-argument generation task aims to automatically generate a statement that has a different stance from the original argument ([Toulmin, 2003](#); [Damer, 2009](#)). Existing works describe it as a paragraph-level generation task ([Hua and Wang, 2018](#); [Alshomary et al., 2021](#); [Alshomary and Wachsmuth, 2023](#)). However, sentence-level counter-argument generation can be quite different. The main challenge of sentence-level generation is to condense the counter-argument into a concise

---
*Corresponding author.
[1]`https://www.reddit.com/r/changemyview/`

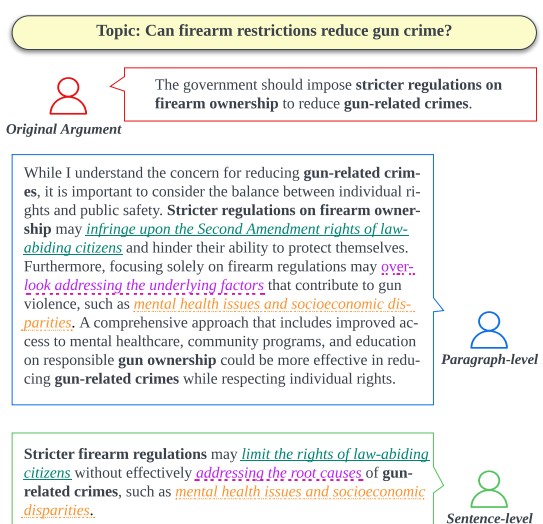

Figure 1: An example that elucidates the difference between paragraph-level and sentence-level counter-arguments. **Topic words** reflecting the discussion points are in bold. Words that are underlined and in the same color denote the key points shared between two counter-arguments.

sentence. It requires identifying the key points and formulating a counter-argument in a limited space. An example of the difference between paragraph-level and sentence-level counter-argument generation is shown in Figure 1.

To address this challenge, we propose a benchmark **ArgTersely** for sentence-level counter-argument generation. The dataset is derived from ChangeMyView (CMV), an online debate forum, and has been annotated by humans.

Recently, large language models, such as OpenAI ChatGPT and GPT-4 ([Bubeck et al., 2023](#)), PaLM ([Chowdhery et al., 2023](#)), and LlaMAs ([Touvron et al., 2023a](#),[b](#)) have achieved great success and demonstrated remarkable performance in text generation tasks. By leveraging the pretrained language model, we propose a framework,

**Arg-LlaMA**, to generate high-quality counter-arguments. Our framework is a pipeline comprising (1) an instruction component, (2) a language model, and (3) a filter component. The instruction component comprises multiple Chain-of-Thought (CoT; Wei et al., 2022) instructions addressing common errors in debates along with their corresponding reasoning steps. As for the language model, we utilize instruct-tuning (Wei et al., 2021) on LlaMA-7b (Touvron et al., 2023a) with the Low-rank Adaptation (Hu et al., 2021) method. During inference, we employ multiple CoT instructions as input for the language model and utilize the filter component to select the best candidate counter-argument as the output of the system.

Previous work typically employed n-gram-based metrics such as BLEU (Papineni et al., 2002) and ROUGE (Lin, 2004) for rapidly evaluating the quality of counter-argument generation (Alshomary et al., 2021; Schiller et al., 2021). However, we believe that these metrics do not effectively assess whether the generated sentences are *pertinent* and *in line with human preferences*. To this end, we propose incorporating model-based metrics, **Arg-Judge**, as a supplementary evaluation approach. Specifically, we trained a BERT-based (Devlin et al., 2019) model, using the human preference data generated during the annotation process. In addition, we introduce a metric, ChatGPT Eval, which we obtain by using ChatGPT to score the sentence's position and argument completion. Moreover, we have made the human evaluation more specific by asking human annotators to assess the outputs based on five dimensions, which enables a comprehensive evaluation of the model's performance.

Our contributions are mainly as follows:

- We propose a benchmark, **ArgTersely**, for sentence-level counter-argument generation and a dataset annotated by humans.
- We propose a counter-argument generation framework, **Arg-LlaMA**. The framework is capable of generating high-quality counter-arguments.
- We propose a novel, lightweight evaluator, **Arg-Judge**, which enables it to reflect the real ranking and is highly consistent with human evaluation.

## 2 ArgTersely

### 2.1 Task Formulation

The task input consists of two components: a topic and an original argument. (1) The topic, denoted as $\tau$, explains the premise of the dialogue and the focus of the debate. (2) The original argument, denoted as $x$, is a sentence containing the initial perspective or stance put forward. The objective of this task is to generate a sentence, $y$, that provides a coherent rebuttal response to $x$ based on the given topic $\tau$.

### 2.2 Dataset Creation

We based our dataset annotation on the CMV dataset (Tan et al., 2016), sourced from the ChangeMyView (CMV) subreddit[2]. CMV users post various topics, with a unique quoting dynamic: User B quotes a segment of User A's statement (usually a sentence) and responds (often with multiple sentences). We extracted $20,626$ triplets from CMV, emphasizing reply relationships. Each triplet includes a topic, a quoted statement (original argument), and its reply. The reply, split into sentences, forms a candidate set. Annotators select sentences from this set that counter the original argument, creating counter-argument pairs. Sentences are skipped if they are incomplete, lack a viewpoint, or breach our ethical guidelines (Appendix F). Any sentence flagged by an annotator for violating these guidelines is excluded. Our annotation process comprises data preprocessing, trial annotation, and formal annotation to ensure dataset quality.

**Data Preprocessing** We segment User B's replies into candidate sentences using punctuation and remove those with hyperlinks, emails, phone numbers, emojis, etc. Grammar issues, such as capitalization and spacing, are corrected.

**Trial Annotation** Before formal annotation, annotators are trained on rules and the annotation system. After that, they undertake trial annotation on 50 triplets. We retained annotators who displayed high consistency with our reference annotations, totaling 24.

**Formal Annotation** To minimize bias, we employ a cross-annotation strategy. Two annotators assess the same triplet, and any discrepancies are resolved by a third. Approximately 30% of the dataset is generated through arbitration.

---

[2]https://www.reddit.com/

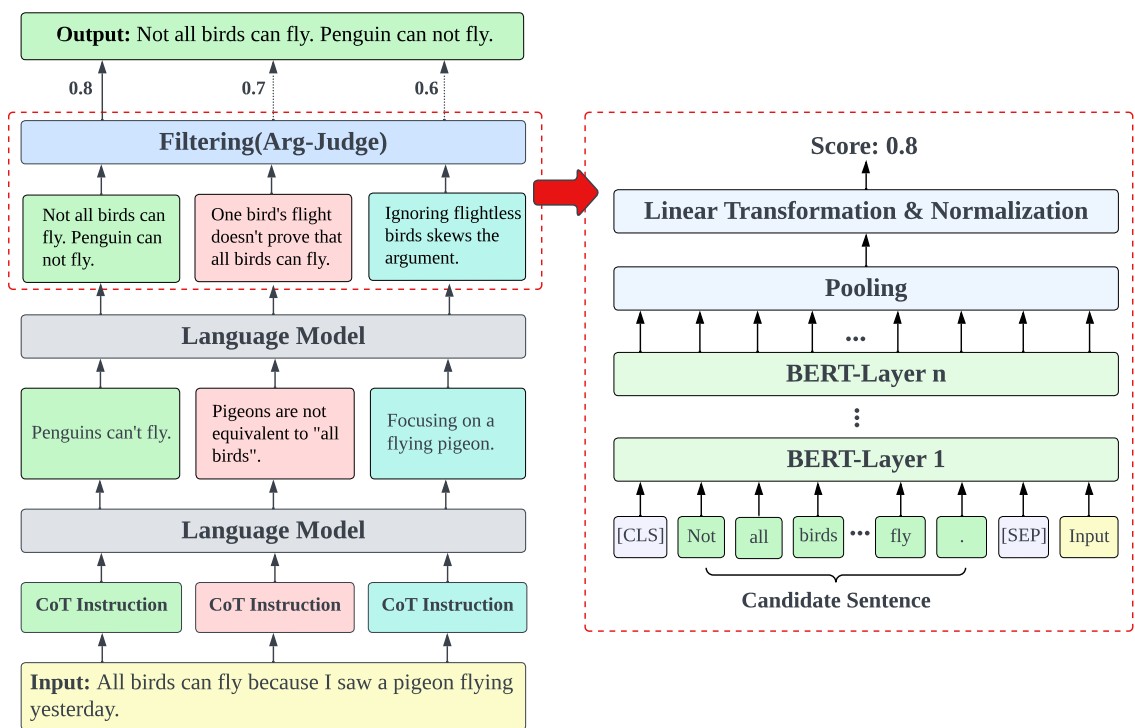

Figure 2: The overview of our proposed framework Arg-LlaMA. First, CoT instructions guide the language model to identify errors. Next, the LM generates candidate sentences based on those errors. Finally, BERT-based filter selects the best counter-argument by scoring the concatenated original argument and candidate sentence.

All annotators have substantial debate experience and at least a bachelor's degree. The annotation process spanned 42 days, yielding 31, 197 argument-counterargument pairs, each associated with the relevant topic. We highlight the ethical considerations during the annotation process, including potential risks, identifiable information, compensation, and annotation biases in Section 9. The statistics of ArgTersely are shown in Table 1.

|  | Train | Valid | Test |
|---|---|---|---|
| # of topic | 7911 | 878 | 2000 |
| # of pair | 28197 | 1000 | 2000 |
| Avg. | | | |
| # words per argument | 21.74 | 21.57 | 19.96 |
| # words per counter-argument | 25.09 | 27.44 | 34.92 |

Table 1: The statistics of ArgTersely.

## 3 Arg-LlaMA

Figure 2 shows the framework we proposed, Arg-LlaMA. It is mainly composed of two parts: 1) a language model (LM) with instruct-tuning, for generating counter-argument, and 2) a filter, for se-

lecting high-quality counter-argument. We employ LoRA and instruct-tuning methods to obtain an LM. Additionally, we leverage human preference data to train the filter.

During inference, we use CoT instructions as inputs of the LM. After obtaining a series of outputs from the LM, the filter will select the best counter-argument as output. The generation pipeline is detailed in Section 3.3.

### 3.1 Instruct-tuning the LlaMA

**Instruction Set Creation** In line with the self-instruct (Wang et al., 2023) approach, we initially generated 148 instructions based on 10 seed instructions. Following a manual verification process, these instructions were expanded to form an Argumentation Instruction Set consisting of 2,772 instructions. Specifically, our specific implementation differs from the self-instruct method in the following aspects:

1. Our seed instructions focus on argument-related instructions, such as "Provide evidence to support the conclusion", "Point out its logical error", etc. A detailed list of specific instructions

is shown in Appendix A and attached files.

2. We use the ChatGPT [3] to generate instructions, enabling us to generate more diverse and elaborate contexts.

**Low-Rank Tuning** Using the above Argumentation Instruction Set and Alpaca instruction set (Taori et al., 2023), we fine-tuned LlaMA-7b model with LoRA method. LoRA maps the weight update of the self-attention module projection matrix in the Transformer (Vaswani et al., 2017) architecture to a lower dimension and then returns to the normal output dimension. In our work, we performed LoRA on all Query/Key/Value/Output projection matrices in the self-attention module.

## 3.2 Training the Filter

The filter component is also a language model. We designed this component with the purpose of selecting high-quality counter-arguments from candidate sentences.

**Ranking Data for training** Our training data, named Ranking Data (RD), originates from human preference data generated during the annotation process of ArgTersely dataset. Given an original argument $x$, we assign ranking scores to candidate sentences based on the following rules:

**1** = Sentences selected by annotators that can form a strong rebuttal relationship with $x$.

**2** = Sentences not selected by the annotator but belonging to the same conversation as $x$.

**3** = Safe reply, randomly selected from a predefined list, as listed in Appendix B

**4** = Sentences sampled from other conversations.

We finally got 20,000 training samples and 800 testing samples, each sample consists of an original argument and four candidates. We denoted the original argument as $x$, the candidates list as $Y = [y_1, y_2, y_3, y_4]$, and the ranking score for $y_i$ as $s_i, i \in \{1, 2, 3, 4\}$.

**Training Task** The training task is learning to rank the candidates in the correct sequence. In this task, we assign the ranking scores of four candidates as the ground truth, with higher scores indicating lower quality.

To optimize the parameters $\theta$ of the filter, we first used the parameters of BERT-base (Devlin et al., 2019) to initialize it. The loss function we

employed is cross-entropy loss:

$$\mathcal{L} = -\log \mathcal{P}_\theta(s_i | x, y_i), i \in \{1, 2, 3, 4\} \quad (1)$$

## 3.3 Generation Pipeline

The generation pipeline consists of three steps: 1) provide CoT instructions to guide the LM, 2) use the LM, generates outputs based on instructions, and 3) apply filtering to refine and obtain the final result by selecting the most appropriate counter-argument.

**CoT Instruction** Our generation pipeline starts with a series of CoT instructions. We propose CoT instructions to guide the model in generating realistic and logical arguments through multi-step reasoning. Based on Kee's (2006) debating theory, we design few-shot and multi-step reasoning templates for several common errors in debate. We roughly divide common errors into the following categories:

- **Factual Error**: a mistake in the presentation of fact.
- **Logical Fallacy**: errors in reasoning that undermine the validity of an argument.
- **Confirmation Bias**: errors in selectively interpreting information in a way that supports existing hypotheses.

The specific formats of these instructions are listed in Appendix C. During inference, we provide the LM with a set of instructions that correspond to the aforementioned errors for generating the counter-arguments.

**LM** The LM serves two roles. Firstly, it acts as an error identifier, tasked with identifying errors within the original argument. Secondly, it generates a candidate counter-argument for each instruction provided.

**Filter** After the LM with various CoT instructions, we get a set of candidate counter-argument $Y = [\hat{y_1}, \hat{y_2}, ..., \hat{y_n}]$. The purpose of the filter is to select the one that maximizes the probability as the output of the system:

$$y^* = \arg\max_{y_i} \mathcal{P}_\theta(s_i = 1 | x, y_i), i = 1, ..., n \quad (2)$$

## 4 Experiments

### 4.1 Experiment Setup

**Dataset** Our experiments are performed on the test set of ArgTersely dataset. It consists of 2,000

---

[3] We use gpt-3.5-turbo as in https://platform.openai.com/docs/models/gpt-3-5, and ditto.

|            | BLEU  | ROUGE | METEOR | ChatGPT Eval | Arg-Judge | # Word | # Param (b) |
|------------|-------|-------|--------|--------------|-----------|--------|-------------|
| BART       | 5.20  | 11.75 | 6.38   | 30.40        | 7.17      | 11     | 0.1         |
| GPT-2      | 13.52 | 15.74 | 14.51  | 31.47        | 23.26     | 15     | 0.8         |
| DialoGPT   | 16.54 | 18.72 | 17.50  | 16.96        | 9.98      | 15     | 0.8         |
| LlaMA      | 15.85 | 22.07 | **21.77** | 35.02     | 37.46     | 27     | 7           |
| Alpaca-LoRA| 15.56 | 18.67 | 17.53  | 46.82        | 47.51     | 36     | 7           |
| GPT-3      | 10.42 | 13.97 | 12.66  | 45.85        | 42.33     | 15     | 175         |
| **Arg-LlaMA** | **18.60** | **22.41** | 19.29 | **50.48** | **55.78** | 38 | 7 |

Table 2: Main result on sentence-level counter-argument generation. We report BLEU-1 (BLEU), ROUGE-L (ROUGE), METEOR, ChatGPT Eval, Arg-Judge, average number of words per sentence(# Word) and the number of parameters in each model(# Param). The best results are in bold. Our proposed model perform well in most metrics (Wilcoxon signed rank test (Kotz and Johnson, 2012), $p < 0.05$).

triplets in the format of <topic, original argument, counter-argument>.

**Implementation Details** When training the base LM with instruct-tuning, we use LlaMA-7b as the base model. We set the learning rate to $3 \times 10^{-4}$, batch size to 256, gradient accumulation step to 16, and train the model 5 epochs on 4 NVIDIA RTX3090 GPUs. The $\alpha$ and $r$ of the LoRA method are both set to 16. When training the filter, we use BERT-base as the base model. We set the learning rate to $1 \times 10^{-5}$, batch size to 64, and train on an NVIDIA RTX3090 GPU for 2 epochs. For training both models, we employed AdamW (Loshchilov and Hutter, 2018) as optimizer.

**Models for Comparision** We compare our system with several baselines:

- BART (Lewis et al., 2020): a pre-trained language model with encoder-decoder structure, and we fine-tuned it to adapt this task.
- GPT-2 (Radford et al., 2019): a pre-trained language model with decoder-only structure.
- DialoGPT (Zhang et al., 2020): a decoder-only language model which was trained on online dialogue corpus.
- LlaMA (Touvron et al., 2023a): a collection of models trained on publicly available datasets, and we use LlaMA-7b.
- Alpaca-LoRA (Taori et al., 2023): a model obtained by LoRA-tuning LlaMA-7b based on the Alpaca instruction set.
- GPT-3 (Brown et al., 2020): a large language model without instruct-tuning.

### 4.2 Evaluation Metrics

Our evaluation metrics include automatic evaluation metrics and human evaluation.

**Automatic Evaluation** First, we do not entirely disregard n-gram-based automatic evaluation metrics that commonly utilized, including BLEU (Papineni et al., 2002), ROUGE (Lin, 2004) and METEOR (Lavie and Agarwal, 2007).

However, of greater importance, we present two *model-based* evaluation metrics to assess performance differences among different systems. A detailed explanation follows:

- **ChatGPT Eval**: We utilize two instructions to guide ChatGPT in generating the stance score ($S_{st}$) and the argument completeness score ($S_{com}$), both of which range from 0 to 100. The instructions we employed are outlined in Appendix D. The stance score assesses whether the original sentence and the generated sentence have opposing stances, while the completeness score gauges the generated counter-argument's caliber, specifically if it makes logical sense. We employ a weighted average of these two scores to get the final score of ChatGPT Eval.

$$S_{gpt} = \lambda S_{st} + (1 - \lambda)S_{com} \qquad (3)$$

, where $\lambda$ is set to 0.5 in our experiments. To reduce the uncertainty of ChatGPT provided, we set the temperature factor to 0.1.

- **Arg-Judge**: In order to ascertain the degree of relevance and informativeness of the generated counter-arguments, we adopt a "reverse validation" approach using the Filter model that was trained in Section 3.2. To this end, we establish Arg-Judge as the metric for evaluating the efficacy of this approach in identifying meaningful counter-arguments that are not mere nonsense safe replies. Specifically, we normalize the average-pooled hidden before the softmax layer of the filter model $\theta$ to get a continuous

predicted score $\hat{s} \in [0, 4]$. We empirically define the Arg-Judge score as

$$S_{aj} = \max\{\frac{1}{9} - \frac{\hat{s}}{36}, 1 - \frac{9}{4}\hat{s}\} \qquad (4)$$

We selected the hyper-parameter setting based on our observation that large language models (such as Alpaca-LoRA and GPT-3) tend to generate sentences with scores concentrated between 0 and 0.8. Arg-Judge can thus enhance the distinguishability for these high-scoring sentences, while still maintaining the monotonicity.

**Human Evaluation** Based on the work of Hua et al. (2019), we conducted a more detailed human evaluation. Five human judges are asked to rate arguments on a Likert scale of 1 (worst) to 5 (best) across 5 dimensions to evaluate the performance of the systems:

- **Grammaticality**: assess whether output adheres to the rules of grammar.
- **Appropriateness**: focus on whether the output is contextually suitable.
- **Content Richness**: reflect the depth of information provided by output.
- **Logic**: measure the rationality of output.
- **Persuasiveness**: show the extent to which readers are persuaded by output.

Additionally, We use Top-1 to represent the proportion of the best output. We emphasize the importance of human evaluation as it provides results that are more aligned with human value, compared to automatic metrics based on n-grams or models.

### 4.3 Main Results

**Automatic Evaluation** As the results listed in Table 2, we have the following conclusions:

- Our model achieves the best results on most of the metrics. The components in our framework help our system to generate more correct and fluent counter-argument.
- Arg-LlaMA and Alpaca-LoRA outperform non-instruct-tuning models like LlaMA and GPT-3. Because non-instruct-tuning models is tend to enhance and extend the original argument, rather than forming a rebuttal relationship with it, which indicates a stance error.
- Comparing n-grams to reference sentences for argument generation tasks is often insufficient. Our proposed model-based metrics, ChatGPT Eval, and Arg-Judge, not only demonstrate consistency with the n-gram metric, but they are

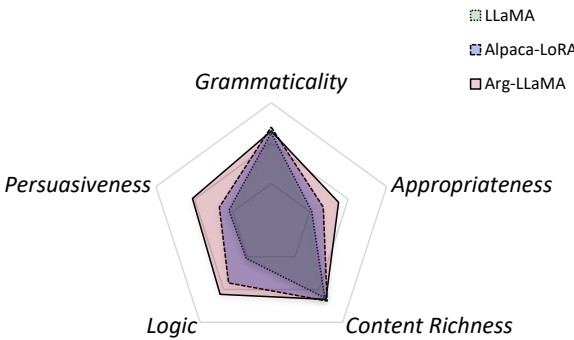

Figure 3: Result of human evaluation on grammaticality, appropriateness, content richness, logic and persuasiveness.

also complemented with each other. For instance, BART and Dialog-GPT models tend to generate stance-correct but logic-and-content-lacking "safe replies", which may receive acceptable scores in BLEU, ROUGE, or ChatGPT Eval, but low scores from Arg-Judge. This addresses the limitations of traditional metrics used in the past.

**Human Evaluation** We report the result of human evaluation in Figure 3 and Table 3, and we have the following observations:

- Our system outperforms in multiple dimensions and Top-1 metric. It shows that the counter-arguments generated by our framework are more in line with human preference.
- Result of Human evaluation is consistent with ChatGPT Eval and Arg-Judge we report in Table 2, which corresponds to our hypothesis that lm-based evaluator may be suitable for counter-argument generation.
- Compared with other models, our system excels in appropriateness, logic, and persuasiveness This achievement can be attributed to the CoT instructions, that effectively guide the language model through multi-step reasoning.

|  | Top-1 |
|---|---|
| Arg-LlaMA | **62%** |
| Alpaca-LoRA | 27% |
| LlaMA | 10% |

Table 3: Result of human evaluation. Top-1 means the proportion of system output ranked as the first. The counter-arguments generated by our proposed model were rated by human judges to be of higher quality.

## 4.4 Ablation Study

We perform ablation studies to explore the role of different components in both training and generation process. We explored four variants in addition to the overall framework: 1) Instead of the argumentation instruction set, we use the Alpaca instruction set to instruct-tuning the LM. 2) We replace the LM with LlaMA-7b, which has not been fine-tuned by instructions. 3) We remove CoT instructions components and use a series of simple instructions, such as "Give me the counter-argument". 4) We remove the filter components and select output from the candidates randomly.

The results of the ablation study are presented in Table 4. We have the following findings:

- **Using argumentation instructions during training is very helpful for the model.** This clearly demonstrates the effectiveness of our proposed argumentative instruction set. We make the argumentation instruction set publicly accessible to benefit the wider community.
- **Instruct-tuning matters.** Simply generating from LlaMA affects performance, while instruction tuning can help the model better adapt to argumentation scenarios, respond to the instructions, and reason out correct rebuttals from CoT instructions.
- **Compared with common instructions, CoT instructions can produce higher-quality counter-arguments.** This is because CoT instructions can give a logical chain and a multi-step reasoning process, which improves the quality of output.
- **Multiple error templates can improve the quality of generated counter-arguments.** Multiple error templates can help the LMs discover potential errors from multiple perspectives, thus generating richer candidate sentences.
- **Filter component plays a crucial role in our system.** It enables us to select high-quality arguments from candidate sentences, while random selection fail to achieve similar performance.

## 5 Validation of Arg-Judge Metric

In order to explore the capability of our proposed Arg-Judge to reflect the actual ranking level and its consistency with human evaluation, we designed two corresponding tasks.

| Models | ChatGPT Eval | Arg-Judge |
|---|---|---|
| *Arg-LlaMA* | **50.48** | **55.78** |
| **Training** | | |
| - argumentation instruction | 44.93 | 51.30 |
| - instruct-tuning | 39.12 | 43.77 |
| **Generation** | | |
| - chain-of-thought | 39.64 | 48.13 |
| - multi-errors | 36.27 | 51.90 |
| - filter | 42.79 | 35.47 |
|    w/ random select | 43.45 | 39.06 |

Table 4: Ablation analysis includes both training and generation modules. ChatGPT Eval and Arg-Judge are both applied.

## 5.1 Datasets for Validation

**Ranking Data (RD)**: We use the test set of this dataset to check if the Arg-Judge can reflect the real ranking. As mentioned in Section 3.2, it has 800 testing samples. A sample includes an original argument and four candidate counter-arguments.

**Quality Selection Dataset (QSD)**: This dataset is used to check whether the Arg-Judge is aligned with human evaluation. It consists of 500 triplets in the format of <original argument, better counter-argument, worse counter-argument>. Given an original arguments from ArgTersely, we first used the ChatGPT to generate two counter-arguments and then manually selected one as the better counter-argument and another as the worse counter-argument.

## 5.2 Validation Tasks and Comparisions

**RD: Can Arg-Judge reflect the real ranking?** Given the original argument, the task on the RD dataset is to select the best counter-argument from four candidates. We use precision at one (P@1) to measure the ability of Arg-Judge to reflect real ranking.

**QSD: Is Arg-Judge consistent with human evaluation?** Given the original argument, the task on the QSD dataset is to select a better counter-argument from two candidates. We use accuracy to reflect the consistency between Arg-Judge and human evaluation.

**Comparisions** We use BERT-base and ChatGPT as comparisons. To adapt ChatGPT to these tasks, we constructed two instructions. Specific information about the instructions is in Appendix E.

## 5.3 Validation Result

**Arg-Judge can reflect the real ranking.** The result in Table 5 shows that the performance of Arg-Judge is better than ChatGPT and BERT. It means that Arg-Judge demonstrates sorting capabilities that reflect real-world scenarios after training.

**Arg-Judge is highly consistent with human evaluation.** Result is in Table 6. Based on the result, the consistency with human evaluation ranks in the following order from high to low: Arg-Judge, ChatGPT, and BERT. It shows the high consistency between Arg-Judge and human evaluation.

|           | P@1  |
|-----------|------|
| BERT      | 0.38 |
| ChatGPT   | 0.53 |
| Arg-Judge | **0.62** |

Table 5: Validation result on the RD dataset. It demonstrates the reliability of sorting the results of Arg-Judge score.

|           | Accuracy |
|-----------|----------|
| BERT      | 0.55     |
| ChatGPT   | 0.77     |
| Arg-Judge | **0.79** |

Table 6: Validation result on the QSD dataset. Arg-Judge shows high consistency with human evaluation.

## 6 Case Study

We illustrate the advantage of our model through a case study in Table 7. Alpaca-LoRA and LlaMA fail to adequately tap into the subtext of the original argument that the white upper middle class cannot help the working class when they are fighting each other. And without instruct-tuning, GPT-3 and LlaMA mostly consist of extensions or additions to the original text, lacking a compelling rebuttal. Whereas, our model recognizes this implicit logic and implements a counter-argument that it is not a plausible excuse.

## 7 Related Work

**Counter-Argument Generation** Datasets (Ji et al., 2021; Yuan et al., 2021; Hua and Wang, 2018; Stab et al., 2018) for counter-argument generation mainly establish the rebuttal relationship in the conversation using automatic methods such as citation or reply detection. Tan et al. (2016)

**Topic:** How should the struggles of white upper middle-class individuals be addressed in the USA?
**Original Argument:** The struggles faced by white upper middle-class individuals in the USA raises questions about the prospects and viability of investing in the white working-class, let alone providing support for communities of color in inner cities.

**Ours:** However, the struggles faced by white upper middle-class individuals in the USA should not be used as an excuse for not investing in the white working-class.

**GPT-3:** The white working-class is not a monolithic group. The white working-class is not a homogeneous group. The white working-class is not a static group.

**Alpaca-LoRA:** The struggles faced by white upper middle-class individuals in the USA raises questions about the prospects and viability of investing in the white working-class and providing support for communities of color in inner cities.

**LlaMA:** Let alone providing support for communities of color in inner cities.

Table 7: Case study of an instance in the test set.

proposed CMV dataset, including the citation relationship between original posts and their corresponding replies. Bolton et al. (2020) introduced Kialo, a dataset for sentence-level argument stance classification and counter-argument generation. ArgTersely distinguishes itself as the first human-annotated dataset of its kind with ranking data reflecting human preferences. Early work (Hua and Wang, 2018; Hua et al., 2019) focus on how to introduce external knowledge into the system; Alshomary et al. (2021) developed a system to identify weak points in arguments; Schiller et al. (2021) developed a controlled argument generation system, which is able to generate arguments based on given information; Alshomary and Wachsmuth (2023) completed it through multi-task and multi-step reasoning. Our work primarily introduces benchmark and evaluation metrics for sentence-level argument generation. Methodologically, we just establish a usable baseline using LLM without introducing too much external knowledge.

**Self-Instruct** A series of recent works (Zhou et al., 2022; Ye et al., 2022; Singh et al., 2022; Honovich et al., 2023) generate instructions of a task given a few examples. Singh et al. (2022) use LLM and reranking algorithm to generate human-interpretable instruction, which matches or even im-

proves upon human-written instruction. Honovich et al. (2023) introduce the instruction induction challenge task and discover the ability to generate instructions emerge when a language model is large enough. Wang et al. (2023) provide an almost annotation-free method for aligning pre-trained language models with instructions. The overall process is an iterative bootstrapping algorithm, which starts off with a limited seed set of manually written instructions that are used to guide the overall generation. We fine-tuned the Arg-LlaMA model using the self-instruct approach, where we included seed instruction for a variety of related tasks of the counter-argument generation.

**Adaptation** As the size of the language model increases, the cost of fine-tuning also increases. A series of works (Shin et al., 2020; Li and Liang, 2021; Houlsby et al., 2019) have studied various methods like prompt-tuning and adapter-tuning to alleviate this problem. However, it's difficult to directly optimize the prompt, and introducing the adapter layer will cause a delay in reasoning. Considering that, Hu et al. (2021) proposed Low-Rank Adaptation (LoRA), which greatly reduces the number of trainable parameters for downstream tasks. Additionally, there have been advancements in extending Low-Rank Adaptation. Dettmers et al. (2023) proposed QLoRA that fine-tunes large models on limited memory GPUs through 4-bit quantization and Low-Rank Adapters. Chen et al. (2023) introduced LongLora that leverages sparse local attention and achieves context extension with minimal computation. Since our work does not involve long contexts in generation and does not prioritize optimization techniques, we just utilize LoRA to fine-tune our model for efficiency.

## 8 Conclusions

In this paper, we introduce a benchmark ArgTersely for sentence-level counter-argument generation. Specifically, we present a human-annotated dataset and develop a language model based on argumentation instructions. We further construct a framework Arg-LlaMA, which leverages the language model. Additionally, we propose two model-based metrics, ChatGPT Eval and Arg-Judge, as complements to n-gram-based metrics. Experiments show that our framework competes well with mainstream models, and our metrics are effective and highly consistent with human evaluations.

## 9 Ethical Considerations

Since we propose a new dataset ArgTersely, we solve some possible ethical issues in this section.

**Potential Risk** Our dataset is sourced from Change-MyView (CMV), a subcommunity on Reddit. Users must adhere to community rules[4], including restrictions on hate speech. We also formulate an ethical guideline and require annotators to follow it. We train annotators to mark and skip sentence violating the ethical guideline. Annotators were informed about potential risks. Our annotation process respects intellectual property and privacy rights.

**Identifiable Information** Our data is sourced from open platforms, safeguarding privacy. We also removed sensitive information such as emails, phone numbers, and usernames during data preprocessing.

**Compensation** We employed 24 part-time annotators, compensating them at $0.25 per conversation (equivalent to at least $3.75 per hour, with a cap of 2 hours per day), which surpasses the local minimum wage.

**Annotation Bias** We perform a series of methods to reduce the bias during annotation, including annotator training, trial annotation, and cross-annotation.

## Limitations

While the experimental results demonstrate the effectiveness of Arg-Judge, it is important to note that our exploration of the consistency between human evaluation and language model evaluators (including ChatGPT Eval and Arg-Judge) was limited to a specific set of scenarios. Furthermore, due to computational resource constraints, we were unable to train a larger-scale language model as an evaluator. Moving forward, our future research will involve expanding the evaluation of the language model evaluator across a broader range of scenarios and utilizing a larger-scale language model to enhance its capabilities.

## Acknowledgments

This work is supported by National Natural Science Foundation of China (No. 6217020551) and Science and Technology Commission of Shanghai Municipality Grant (No.21QA1400600).

---

[4]https://www.reddit.com/r/changemyview/wiki/rules/

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

# A    Examples of Seed Instructions

In Section 3.1, We demonstrated how the instruction set for tuning the LLaMA model was created, and here we present 6 seed instructions, along with examples of input-output pairs. Note: "instruction" and "input" denote the parts we provided as input, whereas "output" denotes the outcomes generated by ChatGPT. Please refer to the *attached file* for all 10 seed instructions.

---

**Instruction:** Give some facts to rebute it.
**Input:** Climate change is not real.
**Output:** The overwhelming majority of scientific evidence supports the fact that climate change is real and is primarily caused by human activities, such as burning fossil fuels and deforestation.

**Instruction:** Point out its logical error.
**Input:** All successful people are happy.
**Output:** It assumes that success is the only factor that determines one's happiness, ignoring other important factors such as personal relationships, mental health, and overall life satisfaction. Success does not guarantee happiness, and happiness can be achieved through various means beyond just professional accomplishments.

**Instruction:** Give evidence to support it.
**Input:** Vaccines are safe and effective in preventing the spread of infectious diseases.
**Output:** Numerous studies have shown that vaccines are safe and effective in preventing the spread of infectious diseases. For example, a study published in the New England Journal of Medicine found that the measles vaccine prevented 95% of cases among vaccinated individuals. Additionally, the World Health Organization reports that vaccines have prevented millions of deaths worldwide from diseases such as polio, measles, and tetanus. The overwhelming scientific consensus is that vaccines are a crucial tool in protecting public health.

**Instruction:** Show evidence that can challenge it.
**Input:** Vaccines are harmful and can cause autism. **Output:** Numerous scientific studies have shown that vaccines are safe and do not cause autism. For example, a study published in the Journal of the American Medical Association found no link between vaccines and autism. Additionally, the Centers for Disease Control and Prevention (CDC) states that there is no evidence to support the claim that vaccines cause autism. The overwhelming consensus among medical professionals is that vaccines are safe and effective in preventing the spread of disease.

**Instruction:** Can we believe this statement?
**Input:** All politicians are corrupt.
**Output:** No, we cannot believe this statement. While there may be instances of corruption in politics, it is unfair and inaccurate to generalize and say that all politicians are corrupt. There are many politicians who work tirelessly for their constituents and uphold high ethical standards in their work. It is important to evaluate individuals based on their actions and not make sweeping generalizations about entire groups of people.

**Instruction:** Identify the potential consequences of following this statement.
**Input:** I don't need to wear a mask, COVID-19 is a hoax.
**Output:** The potential consequences of following this statement include contracting and spreading COVID-19 to others, potentially causing serious illness or death. It also undermines the efforts of healthcare workers and public health officials who are working to control the spread of the virus and protect the community. Additionally, it can lead to fines or other legal consequences for violating mask mandates or public health orders.

# B   The List of Safe Replies

"Safe reply" refers to a direct refutation that is completely unconvincing in an argument. These sentences may potentially be generated through language models, but they are undesirable for argumentation tasks. Therefore, we need to avoid generating these sentences, which include:

- I don't agree.
- I don't think so.
- You are wrong.
- That's incorrect.
- I disagree with you.
- I have a different perspective.
- Your argument is wrong.

# C   Examples of CoT Instructions

---

**Prompt Template**

Following the example in the instruction to generate the counter-argument of input appropriately.
**Instruction:**{Cot Instructions}
**Topic:**{Topic}
**Input:**{Original Argument}
**Counter-argument:**

---

**Factual Error Instruction**

**Example:**
**Input:** {Humans have never set foot on any celestial body other than Earth.}
**Counter-argument:** {This argument has factual error. **1** Astronauts from the USA are humans. **2** Astronauts from the USA had landed on the lunar surface. **3** So humans do have set foot on other celestial body.}

---

**Logical Fallacy Instruction**

**Example:**
**Input:** {If someone is wealthy, they must be highly intelligent.}
**Counter-argument:** {This argument has logical fallacy. **1** The subtext of this argument is that unintelligent people cannot be wealthy. **2** However, intelligence is not the sole determining factor.}

---

**Confirmation Bias Instruction**

**Example:**
**Input:** {All successful entrepreneurs dropped out of college. Therefore, pursuing higher education is unnecessary for achieving business success.}
**Counter-argument:** {This argument has confirmation bias. **1** It disregard the countless successful entrepreneurs who completed their education. **2** So pursuing higher education is still necessary for achieving business success.}

## D Instructions for ChatGPT Eval

**Stance Score**:
Below is a conversation between A and B. Scoring the conversation on a continuous scale from 0 to 100, where score 0 means "B totally support A" and score 100 means "B totally against A".

A: {Original Argument}
B: {Candidate to score}
Score:

- - - - - - - - - - - - - - - - - - - - - - - - - - - - - - - - - - - - - - - - - - - - - - - - - - - - - -

**Argument Completeness Score**:
There is a pair of argument and counter-argument. Given the argument, scoring the counter-argument on a continuous scale from 0 to 100, where score 0 means "really bad counter-argument" and score 100 means "perfect counter-argument".

Argument: {Original Argument}
Counter-Argument: {Counter-Argument}
Score:

## E Instructions for Ranking Data (RD) and Quality Selection Dataset (QSD) tasks

### Ranking Data (RD)

There is an example. Please select a best counter-argument in candidates following the example:

Argument: All birds can fly because they have wings.
1: I don't agree.
2: Not all birds have wings for fly.
3: Pigeons can't fly.
4: Today is Monday.
Answer:2

Argument: {Original Argument}
1: {Candidate1}
2: {Candidate2}
3: {Candidate3}
4: {Candidate4}
Answer:

### Quality Selection Dataset (QSD)

Argument:{Original Argument}
Sentence1:{Candidate1}
Sentence2:{Candidate2}
Given the argument, if I want to select a better counter-argument, I will select sentence

## F   Ethical Guideline in Annotation

Before annotation started, we conducted a round of training for annotators. We train annotators to strictly abide ethical guideline and removed text that violate it, which include:

- Avoid harm to others.
- Be honest and trustworthy.
- Be fair and take action not to discriminate.
- Respect privacy.
- Honor confidentiality.