# OpenReview forum: "Argue with Me Tersely: Towards Sentence-Level Counter-Argument Generation"
_EMNLP/2023/Conference — EMNLP 2023 Main_

### Official Review · Reviewer_C3k2 · 2023-07-31

**Soundness:** 4

**Excitement:**

4: Strong: This paper deepens the understanding of some phenomenon or lowers the barriers to an existing research direction.

**Paper Topic And Main Contributions:**

This paper introduces a new benchmark annotated dataset (ArgTersely) sentence-level counter-argument generation derived from ChangeMyView (CMV), which consists of 31,197 pairs of arguments and their counter-arguments. In addition, the authors of this proposed (Arg-LLaMA) framework are composed of a language model and a filter to select high-quality counter-argument. The goal is to generate high-quality counter-arguments. Additionally, they propose a novel evaluator (Arg-Judge) for the counter-argument generation task.

**Questions For The Authors:**

Q1:  Can you give more information about the annotation process? Did you train the annotators? What about the instructions given to the annotators? In terms of cost, how much time/effort was needed to complete the task? Also, regarding the quality of the annotations, how did you evaluate the annotations?
Q2: If this paper is accepted, will the dataset and the code be made public and ensure the reproducibility?


**Reasons To Accept:**

- well written paper and easy to follow
- new dataset resulting from thorough annotation
- a new framework for generating high-quality counter-arguments


**Reasons To Reject:**

- Regarding the annotation process if you can elaborate more about the question below.


**Reproducibility:**

4: Could mostly reproduce the results, but there may be some variation because of sample variance or minor variations in their interpretation of the protocol or method.

**Reviewer Confidence:**

4: Quite sure. I tried to check the important points carefully. It's unlikely, though conceivable, that I missed something that should affect my ratings.

---

> ### Author Rebuttal · Authors · 2023-08-29
>
> Thank you for taking the time to review our submission. Your thoughtful feedback has been invaluable in refining the quality of our work.
>
> ***Q1: More information about the annotation.***
>
> Please refer to "official comment" for detail.
>
> ***Q2:  Reproducibility.***
>
> We have attached the code in "Supplementary Materials", and will open source this part of the dataset, code and more complete readme in the future.

---

### Official Review · Reviewer_mjoU · 2023-08-03

**Soundness:** 4

**Excitement:**

4: Strong: This paper deepens the understanding of some phenomenon or lowers the barriers to an existing research direction.

**Paper Topic And Main Contributions:**

This paper proposes a new approach to generate sentence-level counter-arguments for a given input argument utilizing Large Language Models (LLMs). To this end, the authors first create a dataset of pairs of claims and their counters from the Change My View (CMV) Subreddit. They then construct a list of instructions relevant to the counter-argument domain, and they find-tune an LLM, namely the llama model, on this dataset accompanied with the instructions. Finally, they use the trained model to generate multiple counters using chain-of-thought (CoT) prompting and select the one that is ranked best according to another bert-based model (called Arg-Judge) that is trained to rank best counter.

To evaluate the approach, the authors use the traditional evaluation measures, the bert-based ranking model (Arg-Judge), and ChatGPT. Their results show that their approach performs better than other baselines, which was also confirmed by the manual evaluation.


**Questions For The Authors:**

- It wasn't clear to me why you don't use the institutions as the prompts for the CoT. Can you elaborate on why this is the case?

- What is the relation between the instructions you created and the critical questions used usually in argumentation schemes?

- Why did you choose to create your own dataset when you could have used the Kialo dataset?

**Reasons To Accept:**

- Novel approach: To my knowledge, it is the first attempt to build an approach that utilizes the LLMs for the task of counter-argument generation

- The resources made available, such as instructions and prompts, are valuable for the community that works on the task

- Extensive amount of experiments to evaluate the claims made in the paper


**Reasons To Reject:**

- There are some minor issues with the papers, but still, no strong reasons to reject them:

	- I found that the creation of the dataset is optional. The Kialo dataset, well-studied in the community, provides exactly what the authors need, pairs of short claims and their counters. It is even cleaner than the dataset the authors created since no automatic processes exist to construct it. Still, what has been created in this paper can be extra data to learn from.

	- The related work, especially regarding counter-argument generation, was shortly laid out with little elaboration about how previous works addressed that task and the implications.

	- In the abstract, the authors claim they trained Arg-Judge with human preference. However, looking at the details of the data that the model is trained on, it turns out that the data is automatically created and does not precisely reflect human preferences.

	- The procedure of creating the seed instructions, expanding them, and mapping them to inputs needed to be clarified. Providing examples here would be very helpful.

**Reproducibility:**

3: Could reproduce the results with some difficulty. The settings of parameters are underspecified or subjectively determined; the training/evaluation data are not widely available.

**Reviewer Confidence:**

4: Quite sure. I tried to check the important points carefully. It's unlikely, though conceivable, that I missed something that should affect my ratings.

**Typos Grammar Style And Presentation Improvements:**

- Line 451: "Arg-Judge is better than the comparison". What does that mean?

---

> ### Author Rebuttal · Authors · 2023-08-29
>
> Thank you for taking the time to review our submission. Your thoughtful feedback has been invaluable in refining the quality of our work.
>
> ***Q1: Why not directly use the Kialo dataset?***
>
> Starting from ArgTersely, we constructed the ranking data based on the rules described in the paper and trained the Arg-Judge with the ranking data. A key to constructing ranking data to train the Arg-Judge is to construct some counter-arguments that can form a rebuttal but "not so good" as negative examples. The Kialo dataset, as an argument relationship classification dataset, provides both PRO and CON labels, but lacks such negative examples. We generated such negative examples during the annotation process, which in turn allow better training of the Arg-Judge.
>
> ***Q2: Lack of introduction to previous work.***
>
> Methods proposed by previous counter-argument generation work are oriented towards the paragraph-level task of generating long-form, logically organized and coherent discourses, which is somewhat different from our proposed task and is introduced in related work section (sec. 7). The sentence-level task we propose encourages the generation of condensed counter-argument, which leads to the fact that the previous model itself can not be well adapted to sentence-level task. After generating a long argument, it needs to be summarized to obtain a streamlined counter-argument. Due to space limitations, the related work section simply lists the work, but does not introduce it in depth. More dataset related work (e.g. the Kialo dataset) will be added in a subsequent version of the submission.
>
> ***Q3: Human preference data is created automatically.***
>
> The ranking data for training Arg-Judge is automatically generated, but the rules are based on the cross-annotation results of the annotators (whether the two annotators tick at the same time), so we believe that the ranking data can reflect human preferences to a great extent. We know that it is difficult to directly label ranking data manually, especially when there are multiple annotators, the inconsistent results are a headache. Therefore, we use this method to indirectly construct ranking data combined with the results of manual annotation.
>
> ***Q4: The procedure of creating Argumentation Instructions.***
>
> We have expanded the instruction set in the way of self-instruct, which has been widely used in recent NLP research. Our seed commands start with 10 human-written commands. The list of seed instructions and prompts for subsequent steps have been attached to the appendix in the paper.
>
> ***Q5: The relationship between Argumentation Instruction and CoT Instruction.***
>
> We hope that the language model can better solve logical problems, which is important for argument mining. So we constructed the argumentation instruction and used it to train the LLM.
>
> In reasoning, we decompose the counter-argument generation task into a multi-step reasoning, first looking for several possible errors from the original argument (such as the factual error, logical error and confirmation bias mentioned in the paper), and then generating counter-argument based on these errors. Our CoT instruction is to better guide the model to complete the reasoning process of finding errors -> generating counter-arguments.
>
> Of course, it does not mean that the two are completely different. There are many CoT instructions such as finding mistakes and generating counter-arguments in the argumentation instruction set we constructed. However, existing research and some of our previous explorations have shown that the richness of instruction will directly affect the zero-shot generalization ability of LLM. In order to obtain a more robust model, our argumentation instruction is designed to be very diverse.
>
> ***Q6: Presentation Improvements.***
>
> We appreciate your diligence in bringing these writing concerns to our attention. Rest assured, we will address and rectify these issues in the upcoming revised version of our submission.

---

### Official Review · Reviewer_ccTv · 2023-08-09

**Soundness:** 3

**Ethical Concerns:**

Yes

**Excitement:**

4: Strong: This paper deepens the understanding of some phenomenon or lowers the barriers to an existing research direction.

**Justification For Ethical Concerns:**

The paper mentioned they proposed a new dataset annotated by human annotators, but neither annotation guidelines, no ethic review permission was mentioned.

**Paper Topic And Main Contributions:**

This paper proposed a new dataset for counter-argument generated, an argument-specific LLM and two new evaluation metrics. The data annotation process needs more description with numbers and guidelines. The framework looks good and complete but needs to compare the more powerful model to prove its performance.

**Questions For The Authors:**

For dataset annotation:
1) How did you measure the human agreement between the annotators?
3) It is really hard to get an exact label for ranking, especially when you have multiple annotators. How did you measure this? (For example, if Annotator 1 decide 1>2>3>4, Annotator 2 decide 2>3>1>4, and Annotator 3 decide 4>3>2>1, how did you decide the final label?)
4) If there are human annotators involved in the annotation task, then you should require ethical review by your institution, but of course, I'm not sure if that applies to your institution.

For model:
1) Have you considered Project Debater from IBM as the baseline? As far as I know, they are one of the most powerful systems for argument mining.
2) How are the constants in Eq. 3 determined?

**Reasons To Accept:**

Good to see the new dataset provided to academia. It will be very helpful for the community to do further research.

Solid work for evaluation including automatic evaluation and human evaluation.

**Reasons To Reject:**

If the dataset is one of the main contributions, then the analysis of the dataset is insufficient. Especially the annotation process of the data, and the consistency between human annotators.

The proposed metrics are not exciting as the author mentioned.

**Reproducibility:**

4: Could mostly reproduce the results, but there may be some variation because of sample variance or minor variations in their interpretation of the protocol or method.

**Reviewer Confidence:**

4: Quite sure. I tried to check the important points carefully. It's unlikely, though conceivable, that I missed something that should affect my ratings.

---

> ### Author Rebuttal · Authors · 2023-08-29
>
> Thank you for taking the time to review our submission. Your thoughtful feedback has been invaluable in refining the quality of our work.
>
> ***Q1: More information about the annotation.***
>
> Please refer to "official comment" for detail.
>
> ***Q2: The measurement of the agreement between the annotators.***
>
> During the annotation of ArgTersely, we compare the consistency of the checking behavior as the measurement of the agreement. For example, one answer is √×√×√, and another answer is √×√√√ , then the consistency ratio is 4/5=80%.
>
> When human evaluation scores a sentence on a scale of 1-5 for a certain aspect, if the two annotators give the same score or only one point difference, we will calculate the average score as the score of the sentence. Otherwise, a third annotator arbitrated and determined the final score. About 15% of the scores are generated by arbitration.
>
> ***Q3: How to get an exact label for ranking from annotation results?***
>
> It is more difficult to directly annotate ranking data manually, especially when there are multiple annotators, the inconsistent results are a headache. Therefore, we use an automatic method to indirectly construct ranking data combined with the results of manual annotation. We introduced the construction method in Sec 3.2 of the paper. Based on the cross-tagging results of the annotators, we only need to compare whether the two annotators have checked/unchecked the sentence at the same time.
>
> ***Q4: Project Debater as baseline.***
>
> Methods proposed by previous counter-argument generation work are oriented towards the paragraph-level task of generating long-form, logically organized and coherent discourses, which is somewhat different from our proposed task. The sentence-level task we propose encourages the generation of short and powerful counter-arguments, which leads to the fact that the previous model itself can not be well adapted to our task (for example, an additional summarization downstream module may need to be added). We will add this baseline in subsequent submissions by explicitly specify character limit.
>
> ***Q5: How to determine the constants in Eq.3?***
>
> This parameter is reflected in the weight of the stance score and the completeness of the argument. We choose 0.5 because we think they are equally important. We tried to evaluate only using stance score or argument completeness score in the early stage. Only using the stance score is not able to distinguish safe replies such as “I don’t agree”. However, using only the argument completeness score can not distinguish the wrong stance of the argument, such as some statements that are complete but support the original argument.
>
> ***Q6: Ethical concerns.***
>
> Please refer to "official comment" to know more about annotator subsidy standards, data acquisition and processing methods, and how we avoid potentially harmful information.
>
> Also, we will add a complete data statement in subsequent submitted papers, including source data introduction, data cleaning method introduction, annotation process introduction and possible potential risks.

---

### Meta-Review · Area_Chair_Mqzj · 2023-09-19

**Recommendation:** 4

**Metareview:**

The reviewers agreed that the novel resource introduced in this paper is an important contribution for the community. However, they also highlighted some drawbacks like the lack of further details on the annotation process and the consistency between human annotators, and the limited discussion and comparison with the related work. Some ethical concerns where also raised about annotators' privacy. The reviewers appreciated the author rebuttal which clarifies most of the issues.

---

### Decision · Program_Chairs · 2023-10-07

**Decision:**

Accept-Main

**Comment:**

The reviewers agreed that the novel resource introduced in this paper is an important contribution for the community. However, they also highlighted some drawbacks like the lack of further details on the annotation process and the consistency between human annotators, and the limited discussion and comparison with the related work. Some ethical concerns where also raised about annotators' privacy. The reviewers appreciated the author rebuttal which clarifies most of the issues.